# Predicting 30-day hospital readmissions using artificial neural networks with medical code embedding

Wenshuo Liu[1], Cooper Stansbury[2,3], Karandeep Singh[1,4,5], Andrew M. Ryan[6], Devraj Sukul [7], Elham Mahmoudi[1,8], Akbar Waljee[1,9,10], Ji Zhu[1,11], Brahmajee K. Nallamothu[1,7,10] *

1 Michigan Integrated Center for Health Analytics and Medical Prediction, University of Michigan, Ann Arbor, MI, United States of America, 2 Department of Computational Biology and Bioinformatics, University of Michigan Medical School, Ann Arbor, MI, United States of America, 3 Department of Systems, Populations, and Leadership, University of Michigan School of Nursing, Ann Arbor, MI, United States of America, 4 Division of Learning and Knowledge Systems, Department of Learning Health Sciences, University of Michigan Medical School, Ann Arbor, MI, United States of America, 5 Division of Nephrology, Department of Internal Medicine, University of Michigan Medical School, Ann Arbor, MI, United States of America, 6 Department of Health Management and Policy, University of Michigan School of Public Health, Ann Arbor, MI, United States of America, 7 Division of Cardiology, Department of Internal Medicine, University of Michigan Medical School, Ann Arbor, MI, United States of America, 8 Department of Family Medicine, University of Michigan Medical School, Ann Arbor, MI, United States of America, 9 Division of Gastroenterology and Hepatology, Department of Internal Medicine, University of Michigan Medical School, Ann Arbor, MI, United States of America, 10 VA Center for Clinical Management Research, VA Ann Arbor Health Care System, Ann Arbor, MI, United States of America, 11 Department of Statistics, University of Michigan, Ann Arbor, MI, United States of America

* bnallamo@med.umich.edu

**Data Availability Statement:** Data are publicly available from the Agency for Healthcare Research and Quality. They may be obtained through a data use agreement with the following site: https://www.

## Abstract

Reducing unplanned readmissions is a major focus of current hospital quality efforts. In order to avoid unfair penalization, administrators and policymakers use prediction models to adjust for the performance of hospitals from healthcare claims data. Regression-based models are a commonly utilized method for such risk-standardization across hospitals; however, these models often suffer in accuracy. In this study we, compare four prediction models for unplanned patient readmission for patients hospitalized with acute myocardial infarction (AMI), congestive health failure (HF), and pneumonia (PNA) within the Nationwide Readmissions Database in 2014. We evaluated hierarchical logistic regression and compared its performance with gradient boosting and two models that utilize artificial neural networks. We show that unsupervised Global Vector for Word Representations embedding representations of administrative claims data combined with artificial neural network classification models improves prediction of 30-day readmission. Our best models increased the AUC for prediction of 30-day readmissions from 0.68 to 0.72 for AMI, 0.60 to 0.64 for HF, and 0.63 to 0.68 for PNA compared to hierarchical logistic regression. Furthermore, risk-standardized hospital readmission rates calculated from our artificial neural network model that employed embeddings led to reclassification of approximately 10% of hospitals across categories of hospital performance. This finding suggests that prediction models that incorporate new methods classify hospitals differently than traditional regression-based

hcup-us.ahrq.gov/nrdoverview.jsp. The statistical
code are freely available with a link provided in the
manuscript.

**Funding:** MIDAS Challenge award Michigan
Institute for Data Science https://midas.umich.edu/
The funders had no role in study design, data
collection and analysis, decision to publish, or
preparation of the manuscript.

**Competing interests:** The authors have declared
that no competing interests exist.

approaches and that their role in assessing hospital performance warrants further investigation.

## Introduction

Approximately 15% of patients discharged after an acute hospitalization are readmitted within 30 days, leading to potentially worse clinical outcomes and billions of dollars in healthcare costs [1]. Given these concerns, multiple quality efforts have been instituted in recent years to reduce readmissions in the United States. For example, the Medicare Hospital Readmission Reduction Program (HRRP) was created as part of the Patient Protection and Affordable Care Act and financially penalizes U.S. hospitals with excess 30-day readmission rates among Medicare beneficiaries [2,3]. Similar programs are being launched for patients with commercial insurance with the goal of further incentivizing hospitals to reduce readmissions [4,5].

Not surprisingly, the development of these programs has led to an increased demand for statistical models that accurately predict readmissions using available healthcare claims data. As the likelihood of readmission is related to key input features of patients (e.g., age and co-morbidities), differences in the distribution of patients across hospitals based on such features may lead to unfair penalization of hospitals that care for more at-risk individuals. Therefore, using statistical prediction models to adjust for patient risk across hospitals is a major priority for accountability programs [6]. However, the performance of prediction models for readmissions have been generally poor. For example, existing methods that rely on regression-based models report area under the curve (AUC) for the receiver operating characteristic in the range of 0.63 to 0.65, suggesting limited discrimination for prediction [7,8]. Recent use of more flexible prediction models that leverage machine learning algorithms, such as random forest and traditional artificial neural network (ANN) models, have attempted to address this limitation with minimal improvements [9–11].

The purpose of this study is to explore whether advances in ANN models and numerical embedding techniques could improve prediction of 30-day readmission using administrative claims data and how this potential improvement may impact calculation of risk-standardized hospital readmission rates. Administrative claims data such as diagnosis code are key to describe a patient's condition and other characteristics, but are often not in the easiest or most straightforward format for statistical analysis. We exploit a word embedding technique classically used in Natural Language Processing (NLP) to convert each diagnosis code into a numerical vector such that the "distance" between diagnosis codes is related to "semantic" similarity. Further, using these numerical vectors as input, we employ a newly developed deep set architecture ANN model to accommodate varying numbers of diagnosis codes across different patients and the fact that the prediction should be invariant with respect to the ordering of the diagnosis codes. ANN models abstract input features from large-scale datasets to assign output probability by approximating a combination of non-linear functions over the input feature-space [12, 13]. Modern deployment of ANN models, including deep learning models, have been used successfully in a range of applications that include image classification and natural language processing [14–17], as well as prediction from electronic heath records [18,19]. We apply embedding algorithms from NLP and a new deep set architecture ANN model to a large United States administrative claims data source focusing on 3 common conditions that were targeted under the HRRP: acute myocardial infarction (AMI), heart failure (HF) and pneumonia (PNA).

## Methods

We conducted this study following the Transparent Reporting of a Multivariable Prediction Model for Individual Prognosis or Diagnosis (TRIPOD) reporting guidelines (see S1 Checklist). All statistical code for replicating these analyses are available on the following GitHub repository: https://github.com/wenshuoliu/DLproj/tree/master/NRD. Data used for these analyses are publicly available at: https://www.hcup-us.ahrq.gov/tech_assist/centdist.jsp.

### Study cohort

We used the 2014 Nationwide Readmissions Database (NRD) developed by the Agency for Healthcare Research and Quality (AHRQ) Healthcare Cost and Utilization Project (HCUP), which includes data on nearly 15 million admissions from 2,048 hospitals [20–22]. The NRD has the advantage of including all payers, including government and commercial insurers. We identified patients hospitalized for AMI, HF, and PNA. We created a separate cohort for each condition using strategies for identifying patients that were adopted from prior published work [8, 23]. The cohort of index admissions for each condition was based on principal *International Classification of Diseases*-9 (ICD-9) diagnosis codes at discharge (e.g. in the case of AMI we used 410.xx, except for 410.x2) while excluding the following cases: (1) records with zero length of stay for AMI patients (n = 4,926) per standards for constructing that cohort (as patients with AMI are unlikely to be discharged the same day); (2) patients who died in the hospital (n = 13,896 for AMI, n = 14,014 for HF, n = 18,648 for PNA); (3) patients who left the hospital against medical advice (n = 2,667 for AMI, 5,753 for HF, n = 5,057 for PNA); (4) patients with hospitalizations and no 30-day follow up (i.e. discharged in December, 2014 (n = 23,998 for AMI, n = 44,264 for HF, n = 47,523 for PNA)); (5) patients transferred to another acute care hospital (n = 8,400 for AMI, n = 5,393 for HF, n = 4,839 for PNA); (6) patients of age < 18 years old at the time of admission (n = 12 for AMI, n = 409 for HF, n = 28,159 for PNA); and (8) patients discharged from hospitals with less than 10 admissions (n = 1,956 for AMI, n = 1,221 for HF, n = 418 for PNA). Given that such facilities (<10 admissions) are not generally considered a part of typical quality assurance or performance measurement programs for readmissions, we were not interested in these facilities. In circumstances where the same patient was admitted several times during the study period, we selected only the first admission. Flow diagrams for the cohort selection are shown in S1 Fig.

### Study variables

Our outcome was 30-day unplanned readmission created using the NRD Planned Readmission Algorithm [23]. The NRD also includes patient-level information on demographics and up to 30 ICD-9 diagnosis codes and 15 procedure codes from each hospitalization. Among the diagnosis codes, the principal diagnosis code at discharge represents the primary reason for the hospitalization while the rest represent comorbidities for the patient. To improve computational efficiency, we only included codes that appeared at least 10 times in the whole NRD database, reducing the number of ICD-9 diagnosis and ICD-9 procedure codes for inclusion in our analyses from 12,233 to 9,778 diagnosis codes and from 3,722 to 3,183 procedure codes, respectively.

### Statistical models and analysis

We evaluated four statistical models: 1) a hierarchical logistic regression model; 2) gradient boosting (using the eXtreme Gradient Boosting [XGBoost] [24] approach, a widely-used, decision tree-based machine learning algorithm) using ICD-9 diagnosis and procedure codes

represented as dummy variables (1 if present, 0 if absent); 3) an ANN model using a feed-for-ward neural network with ICD-9 codes represented as dummy variables; and 4) an ANN model in which ICD-9 codes were represented as latent variables learned through a word embedding algorithm. We used hierarchical logistic regression as a baseline comparator given its ubiquitous use in health services and outcomes research. XGBoost is based on gradient boosted decision trees and it is designed for speed and performance. We used it given its rising popularity in recent years as a flexible machine learning algorithm for structured data. The intuition behind our model comparisons was to explore the differences between sophisticated non-linear statistical models and traditional, "off-the-shelf" machine learning techniques. A more detailed explanation for the statistical models and ANN approaches as well as accompanying statistical code are available in the S1 Information.

To provide a reasonable baseline prediction against which to compare more sophisticated models we constructed a hierarchical logistic regression model trained on account age, gender and co-morbidity data. For co-morbidities, we used the well-established Elixauser Comorbitidy Index [25] to identify 29 variables to include as independent features in the model, with a hospital-specific intercept to account for patient clustering [7]. We implemented this model using the R function glmer from the package *lme4.*

For the second model we trained an XGBoost model on ICD-9 codes and age and gender information in order to provide a comparison to logistic regression. XGBoost has been well-recognized as an "off-the-shelf" ensemble algorithm that extends classical decision trees by iteratively fitting decision trees on the gradient of previous decision trees. XGBoost has been shown to be highly effiecient on large datasets and require little hyper-parameter tuning to achieve state-of-the-art performance in a variety of tasks [26]. We implemented this model using the *Python* package XGBoost with a learning rate of of 0.0002 to prevent potential overfitting.

For the third model we trained a shallow feed-forward ANN on the same set of features as the gradient boosted tree. Our motivation for the ANN architecture was to use a simple design with widely adopted parameters. We employed two fully-connected hidden layers with relu activation functions and a single fully-connected output layer (softmax). We chose the ADAM optimizer with a categorical cross-entropy loss function with a conservative learning rate of 0.0002. We reduced the dimensionality of the input feature space between the fully connected layers from 1,024 to 256 to learn complex patterns from the input features instead of using human-engineered selection of variables (i.e., the Elixhauser Comorbidity Index. ANN models require human parameter specification and may be prone to overfitting. For this reason we kept the architecture of the ANN relatively simple. As such, the ANN model represents a reasonable "off-the-shelf" analogy to the XGBoost model. To further mitigate chances of overfitting we included a dropout layer (0.3). Hyper-parameters were selected through cross-validation to give the best prediction accuracy on a hold-out validation set and evaluated on testing data.

In the fourth model, we encoded 9,778 ICD-9 diagnosis and 3,183 procedure codes into 200- and 50-dimensional latent variable space, using the Global Vector for Word Representations (GloVe) algorithm [27], i.e. each diagnosis code is represented by a 200-dimensional numerical vector and each procedure code is represented by a 50-dimensional numerical vector. We used GloVe, an unsupervised embedding algorithm to project ICD-9 co-occurrences to a numerical feature-space where semantic relations between codes are preserved. The purpose of exploring GloVe embeddings and their potential impact on predictive readmission models is to discover if radical changes from current practices in feature-space and model design impact risk-standardization scores. The prescence of two ICD-9 diagnosis or procedure codes in a patient record during hospitalization was considered as a co-occurrence. We then

counted the number of co-occurrences for each pair of ICD-9 diagnosis and/or procedure codes in the NRD training data, (excluding the testing set) and constructed embedding vectors according to the GloVe algorithm, which uses the global co-occurrence of ICD-9 codes along with a local context. A two-dimensional t-SNE visualization of the embedding vectors of the ICD-9 diagnosis codes is shown in the S2 Fig. The visualization demonstrates that word embedding resulted in related diseases clustering closer to each other and is consistent with the application of word embedding algorithms in other administrative claims data [28, 29].

We used the deep set structure proposed by Zaheer et al [30] to incorporate ICD-9 diagnosis and procedure codes into the ANN model. This allowed us to account for varying counts of secondary ICD-9 diagnosis and procedure codes across patients and allow our model to be invariant to the ordering of these codes (e.g., the $2^{nd}$ and the $10^{th}$ code are interchangeable). The hospital ID was embedded into a 1-dimensional variable–conceptually this is similar to the hospital-level random intercept used in the hierarchical logistic regression models. The architectures of the two ANN models are shown in S3 Fig. The implementation of the ANN models was done using the *Python* packages *Keras* and *Tensorflow*.

To avoid the risk of overfitting, each of the study cohorts were divided into training, validation (for parameter tuning), and final testing sets at a proportion of 80%, 10%, and 10%, stratified by hospitals (i.e., within each hospital). We calculated AUC for the standard hierarchical logistic regression model, the XGBoost model and both ANN models on the final testing set, with the 95% confidence interval given from a 10-fold cross-validation. Once the models were developed, we then calculated risk-standardized hospital readmission rates for both the hierarchical logistic regression and the ANN model trained on diagnosis code embeddings. We calculated these using predictive margin, which is a generalization of risk adjustment that can be applied for both linear and non-linear models (like ANN models) [31, 32]. Specifically, the predictive margin for a hospital is defined as the average predicted readmission rate if everyone in the cohort had been admitted to that hospital. Benefits of predictive margins over conditional approaches have been discussed in Chang et al [33]. We compared this approach to the traditional approach for calculating risk-standardized hospital readmission rates in hierarchical logistic regression models that uses the predicted over expected readmission ratio for each hospital and then multiplying by the overall unadjusted readmission rate [7]; importantly, we found similar results (see S4 Fig).

## Results

### Study cohort

Our study cohort included 202,038 admissions for AMI, 303,233 admissions for HF, and 327,833 admissions for PNA, with unadjusted 30-day readmission rates of 12.0%, 17.7% and 14.3% respectively. The mean (standard deviation) age was 66.8 (13.7) for AMI, 72.5 (14.2) for HF and 69.2 (16.8) for PNA, with the proportion of females 37.6%, 48.9% and 51.8%, respectively. Summary baseline characteristics are shown in Table 1 with additional details of the ICD-9 diagnosis and procedure codes in S1 Table. In these cohorts, we noticed an extremely skewed prevalence of ICD-9 diagnosis and procedure codes that were used to identify features for training related to comorbidities. For example, in the AMI cohort, three quarters of the 5,614 distinct secondary ICD-9 diagnosis codes appear less than 49 times (prevalence 0.02%), while the most frequent ICD-9 diagnosis code (i.e., 41.401 for coronary atherosclerosis of native coronary artery) appears 152,602 times (prevalence 75.5%). See S1 Table for details.

### Performance of prediction models

Results of prediction of 30-day readmission as assessed by AUC are reported in Table 2 for each model and each cohort. The gradient boosting model utilizing XGBoost performed

**Table 1.  Summary statistics of the predictors for each cohort assessed in this study population.**

| | Acute Myocardial Infarction | | Heart Failure | | Pneumonia | |
|---|---|---|---|---|---|---|
| | No Readmission | Readmission | No Readmission | Readmission | No Readmission | Readmission |
| | N = 177,892 | N = 24,146 | N = 249,584 | N = 53,649 | N = 257,135 | N = 46,508 |
| Age, mean (std) | 66.3 (13.7) | 70.5 (13.3) | 72.5 (14.3) | 72.5 (13.9) | 68.6 (17.2) | 70.3 (15.8) |
| Female pct. | 36.60% | 45.00% | 48.80% | 49.30% | 52.60% | 50.20% |
| No. of diagnosis codes, mean (std) | 12.4 (6.1) | 15.7 (6.4) | 15.1 (5.5) | 16.2 (5.7) | 12.7 (5.8) | 14.7 (5.8) |
| No. of procedure codes, mean (std) | 5.6 (3.3) | 5.2 (3.9) | 1.1 (1.9) | 1.3 (2.1) | 0.7 (1.5) | 1.0 (1.8) |

slightly better than the hierarchical logistic regression model and similar to the basic feed-forward ANN model. In general, the medical code embedding deep set architecture model generated the best results on all cohorts relative to the other three models. Compared with hierarchical logistic regression (as a reasonable baseline), the medical code embedding deep set architecture model improved the AUC from 0.68 (95% CI 0.678, 0.683) to 0.72 (95% CI 0.718, 0.722) for the AMI cohort, from 0.60 (95% CI 0.592, 0.597) to 0.64 (95% CI 0.635, 0.639) for the HF cohort, from 0.63 (95% CI 0.624, 0.632) to 0.68 (95% CI 0.678, 0.683) for the PNA cohort. One possible explanation for this performance increase is that the embeddings capture the co-occurrence relationship between diagnosis codes, which is not enjoyed by the other three models, and the ANN is able to learn non-linear mapping patterns between the embeddings and the outcome. In a sensitivity analysis, we repeated the same analysis on elderly patients (65 years old and above) and these are provided in Table 2. Not unexpectedly, the overall AUCs decreased in the sensitivity analysis due to restriction of the cohort by age (which is a powerful predictor of readmission for patients); however, the margins for differences in AUCs across the four different statistical models increased slightly with this restriction by age.

## Risk-standardized hospital readmission rates

Given its overall higher performance, we compared risk-standardized hospital readmission rates calculated from the medical code embedding deep set architecture model with those calculated using the hierarchical logistic regression model. The histograms and summaries of these results are shown in (Fig 1). Distributions of the risk-standardized hospital readmission rates from the two models were similar with just a modest shift downward in the mean for the medical code embedding deep set architecture model. We observed substantial differences in terms of rankings of individual hospitals between the two models. For both models, we divided the hospitals into three groups based on quintiles of predicted risk-standardized hospital readmission rates: top 20%, middle 60% and bottom 20%. For AMI, the medical code embedding deep set architecture model classified 72 (6.4%) hospitals in the middle 60% that the

**Table 2.  Summary statistics of ICD-9CM diagnosis and procedure codes for each cohort.**

| Methods | Acute Myocardial Infarction | Heart Failure | Pneumonia |
|---|---|---|---|
| Hierarchical Logistic Regression | 0.639 (0.635, 0.642) | 0.580 (0.578, 0.583) | 0.605 (0.601, 0.609) |
| XGBoost | 0.666 (0.664, 0.668) | 0.602 (0.599, 0.605) | 0.635 (0.632, 0.638) |
| Feed-Forward Neural Networks | 0.667 (0.664, 0.670) | 0.604 (0.602, 0.606) | 0.639 (0.636, 0.641) |
| Medical Code Embedding Deep Set Architecture | 0.683 (0.680, 0.686) | 0.618 (0.616, 0.621) | 0.656 (0.653, 0.658) |

The prediction accuracy was assessed by the area under the curve for Receiver Operating Characteristic (AUC) on the three cohorts. We compared the four models: the hierarchical logistic regression, XGBoost, the feed-forward neural networks, and the medical code embedding Deep Set architecture model.

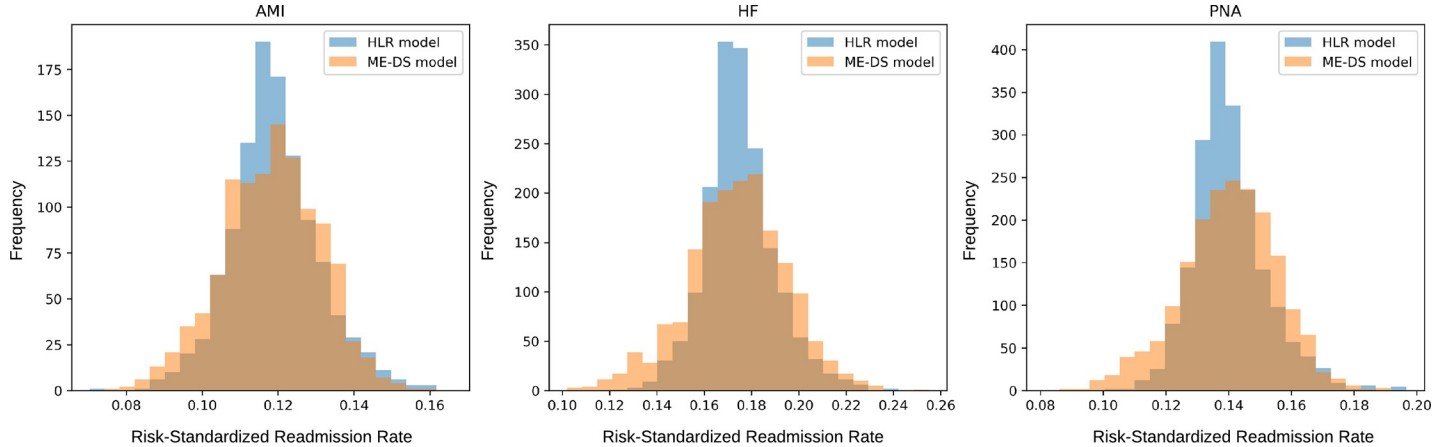

**Fig 1. Distribution of risk-standardized hospital readmission rates.** This figure shows differences in the distribution of risk-standardized hospital readmission rates for acute myocardial infarction (AMI), congestive health failure (HF), and pneumonia (PNA) generated by the hierarchical logistic regression (HLR) model and the medical code embedding Deep Set architecture ANN (ME-DS) model. Standardized readmission rates are generated by comparing model predictions to expected readmission rates for each hospital. This figure illustrates that despite having comparable predictive accuracy LHR and MS-DS lead to differences in hospital risk stratification.

hierarchical model classified in the top 20% and classified 37 (3.3%) hospitals in the middle 60% that the hierarchical model classified in the bottom 20%. Results were similar for the HF and PNA cohorts (Table 3). Given the differences in risk-standardization it is worth investigating whether traditional approaches (like logistic regression) are the best models on which to base risk-standardization scores.

## Discussion

In recent years, ANN models have shown advantages over traditional statistical models in a variety of medical tasks [18, 19]. Whether the application of such models to administrative claims data brings similar improvement in specific tasks related to prediction is worth exploring. This is especially important given the ubiquitous nature of claims data for assessing quality and hospital performance. In this paper, we applied ANN models towards the task of predicting 30-day readmission after AMI, HF, and PNA hospitalizations with and without diagnosis code embeddings. We compared "more sophisticated" statistical models to existing approaches that use input features from classification systems that rely on expert knowledge like hierarchical logistic regression models as well as gradient boosting. Our findings suggest ANN models trained on medical code embeddings provide more accurate predictions of readmission and generate risk-standardized hospital readmission rates that vary from commonly used hierarchical logistic regression models.

**Table 3. Cross tabulation of divided groups between the Hierarchical Logistic Regression (HLR) and the medical code embedding deep set architecture (ME-DS) model for each cohort.**

| | Acute Myocardial Infarction | | | | Heart Failure | | | | Pneumonia | | | |
|---|---|---|---|---|---|---|---|---|---|---|---|---|
| | Rank in HLR model | | | | | | | | | | | |
| Rank in ME-DS model | Top 20% | Middle 60% | Bottom 20% | All | Top 20% | Middle 60% | Bottom 20% | All | Top 20% | Middle 60% | Bottom 20% | All |
| Top 20% | 151 | 72 | 0 | 223 | 235 | 106 | 0 | 341 | 261 | 122 | 0 | 383 |
| Middle 60% | 72 | 563 | 37 | 672 | 106 | 854 | 66 | 1026 | 122 | 949 | 82 | 1153 |
| Bottom 20% | 0 | 37 | 186 | 223 | 0 | 66 | 275 | 341 | 0 | 82 | 301 | 383 |
| All | 223 | 672 | 223 | 1118 | 341 | 1026 | 341 | 1708 | 383 | 1153 | 383 | 1919 |

There has been substantial work performed on constructing risk prediction models to predict readmissions after a hospitalization. The most frequent way these models are employed is through regression-based models that include age, gender and co-morbidities as input features [7]. For co-morbidities, ICD-9 diagnosis and procedure codes obtained from administrative claims data are used as input features to adjust for differences in individual patient risk in these models; however, not all of the thousands of potential ICD-9 diagnosis and procedure codes are included in the models and selecting which to incorporate is an important step. The selection has been based largely on expert input and empirical studies that have been used to generate fixed classification systems like the Hierarchical Condition Categories [34] or Elixhauser Comorbidity Index [25]. Our findings suggest that more attention should be paid to risk-stratification methods based on non-linear classification systems, as they lead to substantial differences in risk-scores.

An advantage of ANN models is their ability as a statistical model to capture potential non-linear effects and interactions of an abstract feature space. By first representing the cooccurance patterns in diagnosis codes using GloVe, followed by a deep set ANN model, one may not need to rely on human-generated classification systems, instead learning to automate extraction of relevant features from the data. Yet few studies to date have explored this type of model towards administrative claims data. We believe a primary reason for this is that ANN models can be difficult to train due to the issues related to parameter optimization and memory consumption in the setting of a large number of parameters–sometimes in the order of millions. In the few studies that have used ANN models with administrative claims data [9, 35, 36], their use also may not have fully captured their full potential for risk prediction. For example, the use of binary "1/0" input features for ICD-9 diagnosis and procedure codes may ignore hidden relationships across comorbidities, limiting the ability of ANN models to improve on traditional hierarchical logistic regression or other methods like gradient boosting.

Of course, there has been some work on predicting readmissions using ANN models in the published literature. Futoma et al. implemented the basic architecture of feed-forward ANN models and showed modest advantages over conventional methods [9]. A number of researchers proposed to embed medical concepts (including but not limited to ICD-9 diagnosis and procedural codes) into a latent variable space to capture their co-relationships [28, 29, 37]; however, these investigators used this approach largely for cohort creation rather than predicting clinical outcomes or risk-adjustment. Krompass et al [36] used Hellinger-distance based principal components analysis [38] to embed ICD-10 codes and then built a logistic regression model using the embedded codes as input features. They found marginal improvements in prediction of readmissions over a feed-forward neural network but were restricted by their limited sample size. Choi et al. [35] designed a graph-based attention model to supplement embedding with medical ontologies for various prediction tasks, including readmission. However, their model did not explicitly consider the fact that the medical codes are permutation invariant. In this paper, we took advantage of a novel word embedding approach, Global Vector for Word Representations (GloVe) [27], as well as a new and recently proposed deep set architecture [30] to fully capture the inter-relationship and the permutation-invariant nature of the diagnosis and procedure codes at a local and global level. These choices–which were purposeful and driven by our intuition on the benefits of ANN models for this specific task–resulted in improved accuracy of prediction for readmission for a word embedding deep set architecture model across all 3 conditions under consideration.

Our study should be interpreted in context of the following limitations. First, although we found ANN models outperformed hierarchical logistic regression models, it is uncertain whether these improvements will justify their use more broadly as this requires consideration of other issues. For example, ANN models require large-scale data sources to train. Even

though such data were available given the NRD for our current work, these are not always available. But the widespread availability and application of administrative claims data in assessing quality and hospital performance justifies the need to explore ANN models (and other approaches) and alternative feature representation further. Second, ANN models are computationally intensive and retain a "blackbox" feel with its findings difficult to understand and explain to users (similar to other models like gradient boosting). These issues may make it less attractive to policymakers and administrators when there may be a need to justify why performance is lacking in a public program (e.g., HRRP). Third, ANN models may not work for applications beyond 30-day readmission in these 3 common conditions. Work is needed to compare the performance of ANN models with traditional approaches for other outcomes (e.g., mortality), rare diseases, or populations (i.e., non-hospitalized patients).

In summary, ANN models with medical code embeddings have higher predictive accuracy for 30-day readmission when compared with hierarchical logistic regression models and gradient boosting. Furthermore, ANN models generate risk-standardized hospital readmission rates that lead to differing assessments of hospital performance when compared to these other approaches. The role of ANN models in clinical and health services research warrants further investigation.

## Supporting information

**S1 Checklist. TRIPOD checklist: Prediction model development.**
(DOCX)

**S1 Fig. Flow diagrams of the cohort creation for Acute Myocardial Infarction (AMI), congestive Health Failure (HF), and pneumonia (PNA).** Cohort selection for AMI, HF, and PNA are shown in (a), (b) and (c) respectively.
(TIFF)

**S2 Fig. Visualization of embedding vectors of the principal diagnosis codes in two dimensions.** This visualization was done using t-SNE. Each point represents a diagnosis code (disease). The size of the points represents the prevalence of that code. (a) The points are coloured by the Clinical Classifications Software (CCS)10 level 1 categories of the multi-level classification system. The frequent codes with the same CCS level 1 categories form clusters, while the infrequent codes form a "cloud" without a clear pattern. (b) As examples, two CCS level 1 categories, "7 Diseases of the circulatory system" and "8 Diseases of the respiratory system" are highlighted in the visualization, with all other diseased represented in grey. (c) The principal diagnosis codes as the inclusion criterion of the three cohorts, acute myocardial infarction, congestive health failure and pneumonia are highlighted.
(TIFF)

**S3 Fig. Architectures of Artificial Neural Network (ANN) models.** (a) Feed-forward neural network ANN model. (b) Medical code embedding deep set architecture model. This model looks up the medical code embedding of each ICD-9 codes that are pretrained by the GloVe model, and aggregates variable number of secondary diagnosis and procedure codes into a final representation vector using the deep set architecture.
(TIFF)

**S4 Fig. Comparison of risk-adjustment methods.** The plots shows the risk-standardized hospital readmission rates and the hospital rankings calculated by the two risk-adjustment methods, predicted over expected readmission rate ratio (e.g., the method employed by the Centers for Medicare & Medicaid Services [CMS]) and predictive margins, for the hierarchical logistic

regression (HLR) model on the acute myocardial infarction cohort.
(TIFF)

**S1 Table. Summary statistics of ICD-9CM diagnosis and procedure codes for each cohort.**
(DOCX)

**S1 Information.**
(DOCX)

## Author Contributions

**Conceptualization:** Wenshuo Liu, Ji Zhu, Brahmajee K. Nallamothu.

**Data curation:** Wenshuo Liu, Cooper Stansbury, Devraj Sukul, Brahmajee K. Nallamothu.

**Formal analysis:** Wenshuo Liu, Cooper Stansbury, Ji Zhu, Brahmajee K. Nallamothu.

**Methodology:** Wenshuo Liu, Ji Zhu.

**Project administration:** Brahmajee K. Nallamothu.

**Supervision:** Brahmajee K. Nallamothu.

**Visualization:** Brahmajee K. Nallamothu.

**Writing – original draft:** Wenshuo Liu, Ji Zhu, Brahmajee K. Nallamothu.

**Writing – review & editing:** Wenshuo Liu, Cooper Stansbury, Karandeep Singh, Andrew M. Ryan, Devraj Sukul, Elham Mahmoudi, Akbar Waljee, Ji Zhu, Brahmajee K. Nallamothu.

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
