## [Decision Letter · Decision Letter 0]

16 Oct 2019

PONE-D-19-22302

Predicting 30-day Hospital Readmissions Using Artificial Neural Networks with Medical Code Embedding

PLOS ONE

Dear Dr. Nallamothu,

Thank you for submitting your manuscript to PLOS ONE. After careful consideration, we feel that it has merit but does not fully meet PLOS ONE’s publication criteria as it currently stands. Therefore, we invite you to submit a revised version of the manuscript that addresses the points raised during the review process.

We would appreciate receiving your revised manuscript by Nov 30 2019 11:59PM. To enhance the reproducibility of your results, we recommend that if applicable you deposit your laboratory protocols in protocols.io, where a protocol can be assigned its own identifier (DOI) such that it can be cited independently in the future. For instructions see: http://journals.plos.org/plosone/s/submission-guidelines#loc-laboratory-protocols

We look forward to receiving your revised manuscript.

Kind regards,

Nan Liu, Ph.D.

Academic Editor

PLOS ONE

Journal Requirements:

Reviewers' comments:

Reviewer's Responses to Questions

**Comments to the Author**

1. Is the manuscript technically sound, and do the data support the conclusions?

Reviewer #1: Yes

Reviewer #2: Partly

2. Has the statistical analysis been performed appropriately and rigorously? 

Reviewer #1: Yes

Reviewer #2: N/A

3. Have the authors made all data underlying the findings in their manuscript fully available?

Reviewer #1: No

Reviewer #2: No

4. Is the manuscript presented in an intelligible fashion and written in standard English?

Reviewer #1: Yes

Reviewer #2: Yes

5. Review Comments to the Author

Reviewer #1: This manuscript (ms) reports an Artificial Neural Network (ANN) based machine learning system for predicting 30-day hospital readmissions. The results show that including medical code embedding improves performance (AUC). The ms is well-written and well-organized.

My concerns are:

1. The exclusion criterion #8 (page 6, line 172) says "with less than 10 admission". Why this was used as a criterion?

2. It would be nice if the authors can provide some explanation why ANN models provide more accurate prediction of readmission and generate risk-standardized hospitalization rates.

3. Table 3 was prepared by grouping the data at top 20%, middle 60%, and bottom 20%. Why these ranges were chosen? Is there any scientific reasoning?

4. Is the improvement in performance significant? The results suggest marginal improvement.

Reviewer #2: There are both pros and cons of the paper:

Pros:

1. The paper compared the performance of 4 different models on the application of predicting unplanned readmission based on a large database. The outcome is accurately and clearly defined.

2. The result of the experiments seems good and it suggests the combination of ANN and medical code embedding as the top method for the readmission prediction.

3. The author presented a very detailed introduction and discussion of ANN and embedding. Different models are well defined.

Cons:

1. Some essential details of the experiments are missing. E.g. What are the parameters of your ANN and Xgboost and how did you tune them. What's the structure of your ANN? How many hidden neurons? What are the features you put in and how they are selected? These parameters are highly related to the performance.

2. The contribution of this work is not well-stated. It looks like that this paper is to recommend a method that combines medical code embedding and ANN for predicting readmission through the comparason. However, the clinical value of the proposed method in readmission prediction is not well-stated. Besides, the novelty of this paper should be addressed.

3. In terms of your results, I don't think there is much difference in the comparison to show the superior of your proposed methods given the unclear experimental details. Suggest you can add other evaluation metrics such as PRC, sensitivity, F1 score.

4. I am also quite confused about your comparisons.

If my understanding is correct, the 4 comparisons in you paper are 1. LR+comorbidity index 2.XGBoost+dummy variables 3.ANN+ dummy variables 4.ANN+ embedding. I would suggest you should add XGBoost+comorbidity index, ANN+comorbidity index, XGBoost+embedding, LR+dummy variables to fully justify the superior of embedding and ANN that you recommended.

6. PLOS authors have the option to publish the peer review history of their article (what does this mean?). If published, this will include your full peer review and any attached files.

Reviewer #1: No

Reviewer #2: No

---

## [Author Response · Author response to Decision Letter 0]

16 Dec 2019

Please see the attached document with detailed responses to the reviewers and editors.

---

## [Decision Letter · Decision Letter 1]

30 Jan 2020

PONE-D-19-22302R1

Predicting 30-day Hospital Readmissions Using Artificial Neural Networks with Medical Code Embedding

PLOS ONE

Dear Dr. Nallamothu,

Thank you for submitting your manuscript to PLOS ONE. After careful consideration, we feel that it has merit but does not fully meet PLOS ONE’s publication criteria as it currently stands. Therefore, we invite you to submit a revised version of the manuscript that addresses the points raised during the review process.

We would appreciate receiving your revised manuscript by Mar 15 2020 11:59PM. To enhance the reproducibility of your results, we recommend that if applicable you deposit your laboratory protocols in protocols.io, where a protocol can be assigned its own identifier (DOI) such that it can be cited independently in the future. For instructions see: http://journals.plos.org/plosone/s/submission-guidelines#loc-laboratory-protocols

We look forward to receiving your revised manuscript.

Kind regards,

Nan Liu, Ph.D.

Academic Editor

PLOS ONE

Reviewers' comments:

Reviewer's Responses to Questions

**Comments to the Author**

1. If the authors have adequately addressed your comments raised in a previous round of review and you feel that this manuscript is now acceptable for publication, you may indicate that here to bypass the “Comments to the Author” section, enter your conflict of interest statement in the “Confidential to Editor” section, and submit your "Accept" recommendation.

Reviewer #1: All comments have been addressed

Reviewer #2: All comments have been addressed

2. Is the manuscript technically sound, and do the data support the conclusions?

Reviewer #1: Yes

Reviewer #2: Yes

3. Has the statistical analysis been performed appropriately and rigorously? 

Reviewer #1: Yes

Reviewer #2: Yes

4. Have the authors made all data underlying the findings in their manuscript fully available?

Reviewer #1: Yes

Reviewer #2: No

5. Is the manuscript presented in an intelligible fashion and written in standard English?

Reviewer #1: Yes

Reviewer #2: Yes

6. Review Comments to the Author

Reviewer #1: All comments have been sufficiently addressed in the revised manuscript. No further comments on the revision.

Reviewer #2: Thank you for addressing my comments. Methods are not very innovative but reasonable. The data and analysis could support the conclusion. The manuscript is well-written.

Some minor problems: there might be typos or document disorder as I cannot find corresponding SFigures in the supporting document (they are SFigure or eFigure?). Authors should recheck it. Besides, I would suggest more explanation/clarification for the Figure 1 (such as adding axis title)

7. PLOS authors have the option to publish the peer review history of their article (what does this mean?). If published, this will include your full peer review and any attached files.

Reviewer #1: No

Reviewer #2: No

---

## [Author Response · Author response to Decision Letter 1]

3 Mar 2020

Please see our responses to the reviewer comments in "[]" below.

REVIEWER COMMENTS

1. If the authors have adequately addressed your comments raised in a previous round of review and you feel that this manuscript is now acceptable for publication, you may indicate that here to bypass the “Comments to the Author” section, enter your conflict of interest statement in the “Confidential to Editor” section, and submit your "Accept" recommendation.

Reviewer #1: All comments have been addressed

Reviewer #2: All comments have been addressed

2. Is the manuscript technically sound, and do the data support the conclusions?

Reviewer #1: Yes

Reviewer #2: Yes

3. Has the statistical analysis been performed appropriately and rigorously?

Reviewer #1: Yes

Reviewer #2: Yes

4. Have the authors made all data underlying the findings in their manuscript fully available?

Reviewer #1: Yes

Reviewer #2: No

[Response: The Nationwide Readmissions Database (NRD) used for this study is a repository from Healthcare Cost and Utilization Project (HCUP), which is maintained by the Agency for Healthcare Research and Quality (AHRQ). We have provided a direct link to the HCUP website in the manuscript where researchers can take steps to access the third-party data used in our analysis. The link was included in all previous submissions as is on (page 4, line 232) of the revision attached:

https://www.hcup-us.ahrq.gov/tech_assist/centdist.jsp

We are bound by a Data Use Agreement with HCUO that forbids us from republishing the data.]

5. Is the manuscript presented in an intelligible fashion and written in standard English?

Reviewer #1: Yes

Reviewer #2: Yes

6. Review Comments to the Author

Reviewer #1: All comments have been sufficiently addressed in the revised manuscript. No further comments on the revision.

Reviewer #2: Thank you for addressing my comments. Methods are not very innovative but reasonable. The data and analysis could support the conclusion. The manuscript is well-written.

Some minor problems: there might be typos or document disorder as I cannot find corresponding SFigures in the supporting document (they are SFigure or eFigure?). Authors should recheck it. Besides, I would suggest more explanation/clarification for the Figure 1 (such as adding axis title)

[Response: We thank the reviewer for a careful review of the manuscript.

Regarding Figure 1. We have added two additional sentences to the caption to guide readers in interpreting distributions of risk-adjusted readmissions rates (page 12, line 502). We have also added axis labels to Figure 1.

Regarding supplemental figures (SFigures): we have included a separate revision of the supplemental material addressing figure labels. All attached figures are correctly labeled in the manuscript and in supplementary material.] 

7. PLOS authors have the option to publish the peer review history of their article (what does this mean?). If published, this will include your full peer review and any attached files.

Do you want your identity to be public for this peer review? For information about this choice, including consent withdrawal, please see our Privacy Policy.

Reviewer #1: No

Reviewer #2: No

---

## [Editor Report · Decision Letter 2]

12 Mar 2020

Predicting 30-day Hospital Readmissions Using Artificial Neural Networks with Medical Code Embedding

PONE-D-19-22302R2

Dear Dr. Nallamothu,

We are pleased to inform you that your manuscript has been judged scientifically suitable for publication and will be formally accepted for publication once it complies with all outstanding technical requirements.

With kind regards,

Nan Liu, Ph.D.

Academic Editor

PLOS ONE

Additional Editor Comments (optional):

Thanks for the revision.
---

## [Editor Report · Acceptance letter]

23 Mar 2020

PONE-D-19-22302R2 

Predicting 30-day hospital readmissions using artificial neural networks with medical code embedding 

Dear Dr. Nallamothu:

I am pleased to inform you that your manuscript has been deemed suitable for publication in PLOS ONE. Congratulations! Your manuscript is now with our production department. 

With kind regards,

on behalf of

Dr. Nan Liu 

Academic Editor

PLOS ONE